# Comparison of Screw-In Forces during Movement of Endodontic Files with Different Geometries, Alloys, and Kinetics

**DOI:** 10.3390/ma12091506

**Published:** 2019-05-08

**Authors:** Sang Won Kwak, Chan-Joo Lee, Sung Kyo Kim, Hyeon-Cheol Kim, Jung-Hong Ha

**Affiliations:** 1Department of Conservative Dentistry, School of Dentistry, Kyungpook National University, Daegu 41940, Korea; endokwak@pusan.ac.kr (S.W.K.); skykim@knu.ac.kr (S.K.K.); 2Department of Conservative Dentistry, School of Dentistry, Dental Research Institute, Pusan National University, Yangsan 50612, Korea; golddent@hotmail.com; 3Dongnam Regional Division, Korea Institute of Industrial Technology, Jinju 52845, Korea; cjlee80@kitech.re.kr

**Keywords:** endodontics, engine-driven NiTi file, root canal preparation, root canal therapy, screw-in force

## Abstract

This study compared the maximum screw-in forces of various instruments during their movements. Forty simulated canals in resin blocks were randomly divided into four groups (n = 10): ProTaper Universal F2, ProTaper Gold F2, WaveOne Primary, and WaveOne Gold Primary. To standardize a lumen size, all artificial canals were prepared with ProTaper Universal F1. The rotation speed was set at 350 rpm with an automated 4 mm pecking motion at a speed of 1 mm/s. The pecking depth was increased by 1 mm for each pecking motion until the file reached the working length. During instrumentation, screw-in forces were automatically recorded by customized software. Maximum screw-in forces were analyzed by one-way ANOVA and Tukey’s post hoc comparison with the significance level at 0.05. WaveOne Gold files generated the lowest maximum screw-in forces, followed by ProTaper Gold, WaveOne, and ProTaper Universal (*p* < 0.05). Under the condition of this study, heat-treated nickel–titanium (NiTi) files with smaller cross-sectional area, fewer contact points, and reciprocating movements resulted in a lower screw-in effect.

## 1. Introduction

Because of superior flexibility and higher torsional fracture resistance of nickel–titanium (NiTi) instruments compared to those of stainless-steel (SS) instruments, NiTi instruments are widely chosen for the root canal treatment [1]. Shaping the root canals by using NiTi instruments may allow better preservation of the root canal anatomy and produce fewer procedural errors such as ledge, transportation, and perforation [2,3].

Geometric features and heat treatment could influence the mechanical properties [4,5,6] and clinical performances [6,7,8,9,10,11] of NiTi instruments such as the incidence of file separation and stress distribution through the dentinal wall during root canal shaping. As NiTi instruments make contact with the dentinal wall to cut root dentin during root canal shaping, stress may be generated inside the NiTi instrument and a reactive force may also be generated toward the root dentin. The fluted cutting blades of NiTi instruments have a spiral form in the longitudinal aspect. This helical configuration is essentially necessary for the cutting and removal of infected dentin, but this may cause an unwanted apical driving power (the ”screw-in” effect). The screw-in effect is clinically defined as the tactile sensation of the instrument within the operator’s hand being pulled into the root canal apically. Uncontrolled screw-in forces may bring an unintended overextension of the instrument beyond the apical foramen [12]. Accidental overpreparation caused by screw-in forces may create apical root cracks and result in the weakening of the root structure [13]. Therefore, it is recommended to manipulate the handpiece deliberately and/or apply a brushing motion to avert the instrument from sudden pulling forces into the root canal.

This screw-in phenomenon has been reported to occur frequently when the cutting tip of the rotary instruments are active and have sharp fluted edges [12,14]. Over-manipulation of the instrument beyond the apical foramen by screw-in forces may result in unintended larger apical preparation size at working length and apical perforation. In addition, the sudden onset of pulling forces into the root canal may result in a momentary increase of stress generation on the NiTi instrument, which may be related to the incidence of torsional fracture [15,16]. Consequently, a separated instrument remains in the root canal and/or over-instrumentation beyond the apical foramen may reduce the success rate of endodontic treatment [17,18].

Various kinds of engine-driven NiTi instrument systems made from different alloys and geometric designs are marketed in contemporary endodontics. Few studies have examined the relationship between the features of NiTi files and the screw-in tendency. Previous studies have demonstrated that the screw-in tendency is affected by cross-sectional design and pitch length rather than shaping motion [19,20]. The heat treatment of NiTi alloys is known to improve their flexibility and mechanical performance [21]. Thus, a change in screw-in forces due to the difference in heat treatment may occur. On the other hand, in terms of the kinematic movements, reciprocating instruments were claimed to have reduced screw-in forces because of their periodic rotation direction change [22].

Therefore, the aim of this study was to compare the screw-in forces of various instruments with different heat treatments and kinetics during their movements. The null hypothesis was that the heat treatment, geometric properties, and kinetics of endodontic files will not influence the screw-in force during simulated root canal preparation.

## 2. Materials and Methods

Four file systems made of different alloys and with different kinetic movements were tested: Two continuous rotation systems (ProTaper Universal F2 (PTU; Dentsply Sirona, Ballaigues, Switzerland) and ProTaper Gold F2 (PTG; Dentsply Sirona) and two reciprocating systems (WaveOne Primary (WOP; Dentsply Sirona) and WaveOne Gold Primary (WOG; Dentsply Sirona)). PTU is made of conventional NiTi alloy; WOP is made by using M-wire; PTG and WOG are made with “Gold-Wire” (brand name of heat-treated alloy). All instruments have a tip size of ISO #25 with a different apical taper: PTU, PTG, and WOP have 8% apical taper; WOG has 7% apical taper. Their cross-sectional area at D4, helical angle, and pitch length were measured using 30× magnification under a stereomicroscope (MZ16FA; Leica, Heerbrugg, Switzerland) (Figure 1a–d).

Forty simulated J-shaped root canals in resin blocks (Dentsply Sirona) were used in this study (n = 10). The curvature of root canal was determined to 35° according to the Schneider method [23], and the working length was measured at 16 mm, which is 0.5 mm short from the canal exit under a stereomicroscope (Leica S6D; Leica Microsystems, Wetzlar, Germany). Before measuring the screw-in forces with designated files, a glide path was prepared with ProGlider (Dentsply Sirona) and the canals were pre-enlarged in a sequence of ProTaper Universal S1, S2, and F1. During the instrumentation, canals were irrigated with distilled water. Apical patency was checked by using a #10 K-file (M access; Dentsply Sirona).

A custom-made test device (DMJ system, Busan, Korea) (Figure 2) was fabricated to measure screw-in forces during each file movement. Each resin block and designated file were connected to the device (Figure 2). The rotational speed was set at 350 rpm for all tested files and the reciprocating angles were set as 170° counterclockwise and 50° clockwise to give the same kinetic condition for three reciprocating files. The files were automatically moved to the working length (16 mm) with a pecking motion of a 4 mm distance and the crosshead speed was set at 1 mm/s. The pecking depth was increased by 1 mm for every single pecking motion until the file reached the working length. The screw-in forces generated during the file movement were automatically recorded by using customized software, and the maximum force was extracted from the data.

The maximum screw-in forces during the instrumentation procedures were checked for the normality of distribution using the Kolmogorov–Smirnov test. Then, the data were analyzed using the one-way analysis of variance and Tukey’s post hoc comparison test. The level of significance was set at *p* < 0.05. All statistical analyses were performed using SPSS version 22 for Windows (IBM Corp., Somers, NY, USA).

## 3. Results

The mean and standard deviation values of screw-in forces for different tested endodontic files are shown in Table 1. The geometric characteristics of the tested instruments are presented in Figure 1. A representative strip-chart shown in Figure 3 illustrates the use of WaveOne Gold Primary during all simulated root canal preparation. WOG files showed the lowest maximum screw-in force by a significant amount, as compared to the other tested instruments, followed by PTG, WOP, and PTU (*p* < 0.05).

## 4. Discussion

Previous studies have reported that the geometric features and metallurgical properties of the NiTi file systems have an effect on their mechanical performances [21,24,25,26,27]. These factors may also be related to the apical forces applied to root dentin for root canal shaping [19,20,28]. In this study, the generation of screw-in forces was evaluated in various NiTi instruments with different kinematics, alloys, and designs. PTU is made of conventional NiTi alloy and has an active cutting edge with a convex triangular cross-section (for the size F2; Figure 1c). PTG shares the same cross-sectional design with PTU, but this file system is made of “Gold-Wire” and has some characteristics similar to those of Controlled Memory wire (CM-wire). WOP is made of M-wire and has three points of contact with the canal wall. WOG, a next-generation instrument after WOP, is made of Gold-Wire and has a parallelogram cross-section and minimal contact points (one or two) depending on the canal shape.

The resin blocks with simulated root canals were used for the standardization of experimental condition and might have a methodological limitation. Even though experimental conditions were standardized by the use of simulated resin canals, the properties of resin material are different from those of natural dentin. Although extracted natural teeth could be used to simulate a clinical condition, the screw-in forces generated during root canal shaping may be highly affected by the anatomic deviations such as canal curvature and lumen size shown in natural teeth [19]. A previous study confirmed that there was a corresponding increase in screw-in forces with increased root canal curvature regardless of the cross-sectional design and the number of threads on the NiTi instruments. Under the non-standardized conditions with the natural teeth, the contact area between the canal and instrument may have wide deviation due to the different canal lumen size.

Both the geometric design and manufacturing process (heat treatment) of instruments may affect the flexibility and the reaction forces as well as the screw-in effect [19]. In the present study, WOG produced the lowest maximum screw-in force among all the instruments tested. A previous study, using finite element analysis, revealed that the higher flexibility and a smaller diameter of the instrument resulted in lower screw-in forces during reproducible standardized up-and-down movements in a curved canal [19]. NiTi instruments with a shorter pitch and more threads also resulted in the generation of lower screw-in forces [19,28]. In a geometric aspect, slender rectangular cross-sectional instruments showed lower screw-in forces than triangular and square cross-sectional instruments [6,8,29]. The WOG has the smallest cross-sectional area among all tested instruments (157,200 µm^2^ at a 4 mm level) and a unique cross-sectional geometry (parallelogram) similar to a slender rectangular and one- or two-point contacts with the dentinal wall during root canal preparation. These geometric features of WOG might have contributed to generating the lowest screw-in forces during instrumentation.

In contrast, PTU showed the highest screw-in forces. PTU is made of conventional NiTi alloy and has a triangular cross-sectional area designed to cut root dentin with three-point symmetrical contacts. PTU also has a larger taper (0.08 taper for F2) compared with WOG (0.07 taper for Primary). These characteristics increase its rigidity and may produce a higher screw-in effect.

In the present study, PTG showed the lower maximum screw-in force compared with PTU, despite these two file systems having similar geometric features. This might be explained by the characteristics of Gold-Wire, which might share thermomechanical properties with CM-wire, and cause the least amount of canal transportation due to its higher flexibility and minimized reaction force [30,31]. The manufacturer claims that NiTi files manufactured using the Gold-Wire heat treatment method have increased flexibility compared to the files made of M-wire or conventional NiTi files, thus producing minimal reaction forces on the canal wall [32].

Manufacturers have claimed that reciprocating movements could reduce the screw-in effect [22]. A transitory clockwise rotation (the opposite direction to the active direction) is sufficient to relieve the stress when the instrument is trapped in dentin during counterclockwise rotation. In the present study, PTU (continuous rotating) showed a higher screw-in effect than WOP (reciprocating), and PTG (continuous rotating) generated a higher screw-in force compared with WOG. This might imply that the difference in kinetic movement could influence the decreased screw-in effect. Furthermore, WOP showed a lower screw-in force than PTU, but higher than PTG. The heat treatment and geometric characteristics have a greater effect on generating screw-in forces than shaping motion, because PTU, PTG, and WOG share similar geometric characteristics (taper, cross-sectional area).

Although the screw-in tendency is harmful to root canal preparation, to date few studies have reported the screw-in effect related to NiTi instruments [14,19,20]. Clinicians should be aware of the occurrence of screw-in forces during the root canal shaping procedure. To prevent the instrument from being pulled inside the root canal by generated screw-in forces, holding the hand-piece firmly and using brushing actions would be helpful. Clinicians also need to be aware that geometrical and metallurgical characteristics may influence some instruments that are more prone to high screw-in forces.

## 5. Conclusions

Under the limitations of this study, heat-treated NiTi files (WOG and PTG) generated lower screw-in forces compared to M-wire (WOP) and conventional NiTi files (PTU). WOG, which has a smaller cross-sectional area with fewer contact points, generated lower screw-in forces. Reciprocating movement adopted by WOP and WOG may also have effect on the decreased screw-in effect.

## Figures and Tables

**Figure 1 materials-12-01506-f001:**
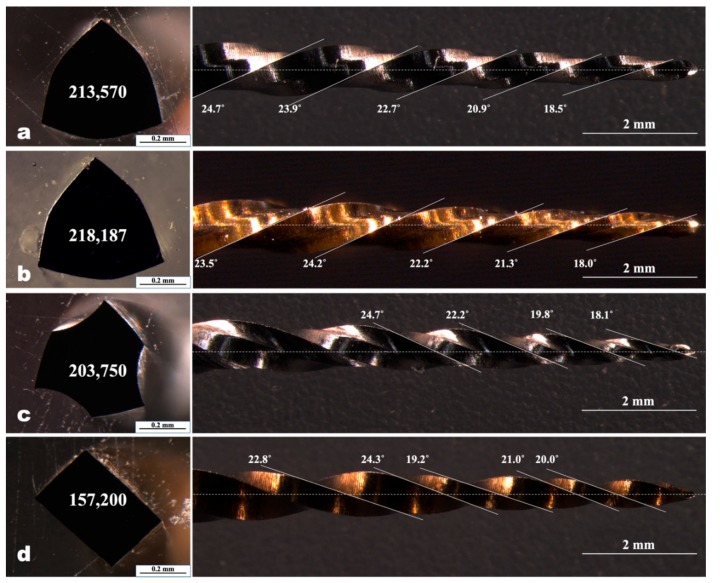
Cross-sectional shape and area (µm^2^) at the level of D4 (left), and longitudinal geometry showing helical angle and pitch (right). (**a**) ProTaper Universal F2; (**b**) ProTaper Gold F2; (**c**) WaveOne Primary; (**d**) WaveOne Gold Primary.

**Figure 2 materials-12-01506-f002:**
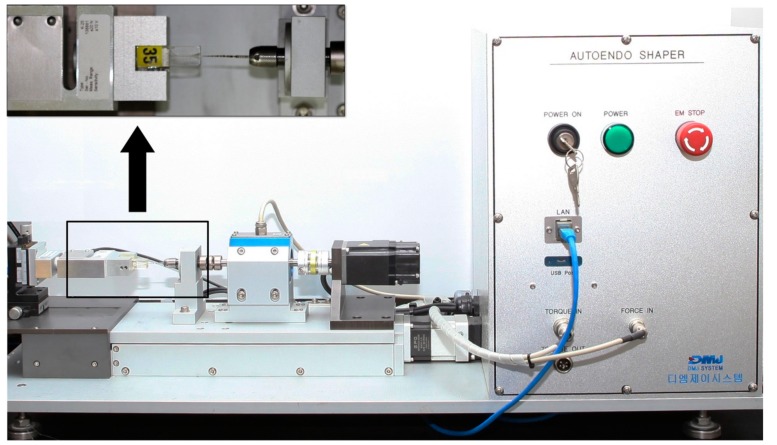
A custom-made test device (DMJ system, Busan, Korea). The simulated canal, force sensor, and engine-driven motor unit are shown in box.

**Figure 3 materials-12-01506-f003:**
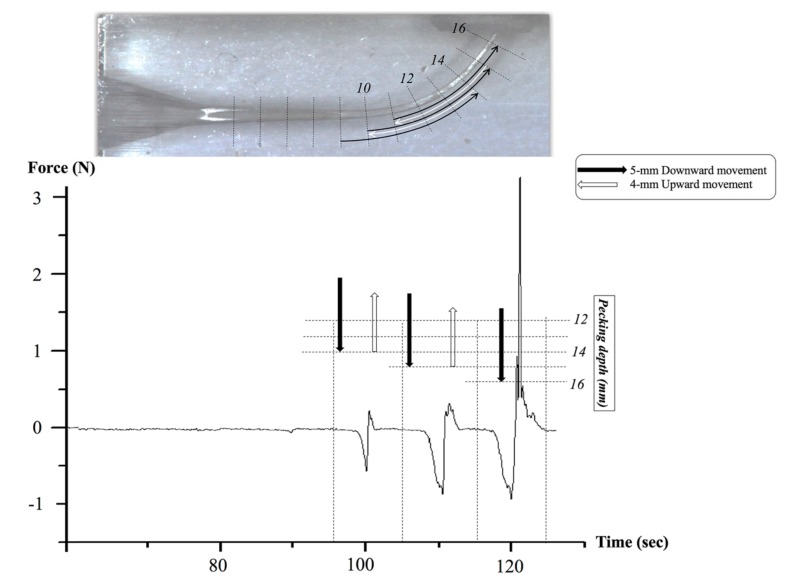
Representative strip-chart of screw-in forces during instrumentation using WaveOne Gold Primary. Negative and positive values indicate the screw-in and apical driving forces, respectively. Repeating 5 mm downward (black arrow) and 4 mm upward (white arrow) movement, the files approach the working length (16 mm) over time. The incremental penetration depth at each stroke was 1 mm.

**Table 1 materials-12-01506-t001:** Maximum screw-in forces (N) during instrumentation procedures.

Group	Mean ± SD
ProTaper Universal, F2 (PTU)	1.941 ± 0.290 ^a^
ProTaper Gold, F2 (PTG)	1.234 ± 0.246 ^c^
WaveOne, Primary (WOP)	1.622 ± 0.130 ^b^
WaveOne Gold, Primary (WOG)	0.976 ± 0.062 ^d^

^a,b,c,d^: Different superscript letters indicate significant differences among groups (*p* < 0.05).

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
