# Peer review of "Comparison of Screw-In Forces during Movement of Endodontic Files with Different Geometries, Alloys, and Kinetics"

_materials, 2019, doi:10.3390/ma12091506_

Reviewer 1 Report

In this study, screw in forces were investigated and interpreted in dependence of motion and metallurgy. The topic is of interest for the limited readership of practical Endodontists and worth publication. However, the authors chose only one file system per group, which does not allow the overall conclusions made throughout the manuscript.

- "significance level of 95%" is definitely an inadequate expression in the abstract, please correct.

- The last paragraph in the discussion section should be place in the introduction to illustrate the gap of knowledge and make an argument for the study.

- Please limit the conclusions made in the text to the systems under investigation.

Author Response

In this study, screw in forces were investigated and interpreted in dependence of motion and metallurgy. The topic is of interest for the limited readership of practical Endodontists and worth publication. However, the authors chose only one file system per group, which does not allow the overall conclusions made throughout the manuscript.

Authors’ response: We rephrased the conclusion part in the revised manuscript under the limitation in system used in the present study. 

 - "significance level of 95%" is definitely an inadequate expression in the abstract, please correct.

Authors’ response: Thank you for the kindness. The manuscript has been revised as the reviewer pointed out.

 - The last paragraph in the discussion section should be place in the introduction to illustrate the gap of knowledge and make an argument for the study.

Authors’ response: We have already mentioned similar contents in introduction section. We added the last paragraph to suggest clinical handling method to reduce screw in force based on the result of this study. 

 - Please limit the conclusions made in the text to the systems under investigation.

Authors’ response: Thank you for the point. The manuscript has been revised 

Reviewer 2 Report

Dear Authors,

 The aim of this study was to compare the screw-in force of four instruments during their movements. Conditions resulted in lower screw-in effect will be useful information for readers in Materials. While the topic is fitting to the journal scope, some minor concerns were raised. Revise the manuscript by following comments.

 Minor points

 Abstract

The authors stated that ProTaper Universal files generated the highest maximum screw-in forces, followed by WaveOne, ProTaper Gold, and WaveOne Gold (p<0.05). In terms of the best instruments, the lowest maximum screw-in forces should be explained in the first order.

 Figure 1

The unit of cross-sectional area (µm2) must be explained in the figure caption.

 The order of listed instruments should be followed the explanation in Materials and Methods.

(a)   ProTaper Universal, (b) ProTaper Gold, (c) WaveOne, (d) WaveOne Gold

 Table 1 “between groups” must be modified to “among groups”.

Author Response

Minor points

Abstract

The authors stated that ProTaper Universal files generated the highest maximum screw-in forces, followed by WaveOne, ProTaper Gold, and WaveOne Gold (p<0.05). In terms of the best instruments, the lowest maximum screw-in forces should be explained in the first order.

Authors’ response: Thank you for the kindness. The manuscript has been revised as the reviewer pointed out.

Figure 1 The unit of cross-sectional area (µm2) must be explained in the figure caption.

Authors’ response: The unit for the cross-sectional area (µm2) has been added in the figure1 legend.

 The order of listed instruments should be followed the explanation in Materials and Methods.

(a)   ProTaper Universal, (b) ProTaper Gold, (c) WaveOne, (d) WaveOne Gold

Authors’ response: Thank for kind mention. We realigned the order of figure1 in the revised manuscript as your recommendation.

 Table 1“between groups” must be modified to “among groups”.

Authors’ response: Thank you for the point. The manuscript has been revised.

Reviewer 3 Report

Congratulatios for the study, comparing the screw-in forces of various endodontic instruments during their movements.

The research is interesting for the clinic, however it has some weaknesses.

Author Response

This study compared the screw-in forces of various endodontic instruments during their movements, using y simulated canals in resin blocks. The research is interesting for the clinic, however it has some weaknesses:

All the mechanized instruments screw into acrylic channels, not comparable with natural teeth;

Authors’ response: We concede that the methodological limitation using resin blocks instead of natural teeth. The hardness of resin block is different from that of dentin. However, as we described in discussion, we decided to use resin blocks for the standardization of experimental condition. Because the contact area between the canal and instrument may have wide deviation due to the different canal lumen size shown in natural teeth and this might have effect on the screw-in force generated.

 The authors did not use the protocol suggested by the manufacturer; The file that less screwed was the one that had less mass, consequently less contact with the walls of the acrylic

Authors’ response: Thank you for the point. The purpose of this study did not simply focus on comparing the screw-in forces from 4 different NiTi files, but the screw-in force generated depending on the cross-sectional areas, alloys, and kinetics. For standardization, we prepared the simulated canal with sequential usage from #10K-file up to PTU F1, which is equivalent to #20 size.

After that, the canal is ready to accept the tested system’s instrument having tip size of #25.  

Round  2

Reviewer 3 Report

Not applicable.